

# Transcriptome analysis of the induction of somatic embryogenesis in *Coffea canephora* and the participation of ARF and Aux/IAA genes

Ana O. Quintana-Escobar[1], Geovanny I. Nic-Can[2], Rosa María Galaz Avalos[1], Víctor M. Loyola-Vargas[1] and Elsa Gongora-Castillo[3]

[1] Unidad de Bioquímica y Biologia Molecular de Plantas, Centro de Investigación Científica de Yucatán, Mérida, Yucatán, México
[2] CONACYT Research Fellow-Facultad de Ingeniería Química, Universidad Autónoma de Yucatán, Mérida, Yucatán, Mexico
[3] CONACYT Research Fellow-Unidad de Biotecnología, Centro de Investigación Científica de Yucatán, Mérida, Yucatán, México

## ABSTRACT

**Background**. Somatic embryogenesis (SE) is a useful biotechnological tool to study the morpho-physiological, biochemical and molecular processes during the development of *Coffea canephora*. Plant growth regulators (PGR) play a key role during cell differentiation in SE. The Auxin-response-factor (ARF) and Auxin/Indole-3-acetic acid (Aux/IAA) are fundamental components involved in the signaling of the IAA. The IAA signaling pathway activates or represses the expression of genes responsive to auxins during the embryogenic transition of the somatic cells. The growing development of new generation sequencing technologies (NGS), as well as bioinformatics tools, has allowed us to broaden the landscape of SE study of various plant species and identify the genes directly involved.

**Methods**. Analysis of transcriptome expression profiles of the *C. canephora* genome and the identification of a particular set of differentially expressed genes (DEG) during SE are described in this study.

**Results**. A total of eight ARF and seven Aux/IAA differentially expressed genes were identified during the different stages of the SE induction process. The quantitative expression analysis showed that ARF18 and ARF5 genes are highly expressed after 21 days of the SE induction, while Aux/IAA7 and Aux/IAA12 genes are repressed.

**Discussion**. The results of this study allow a better understanding of the genes involved in the auxin signaling pathway as well as their expression profiles during the SE process.

## INTRODUCTION

Coffee is one of the most important crops worldwide. It is cultivated in more than 80 countries, occupying 11 million ha in Africa, Asia, and America (*Denoeud et al., 2014*; *ICO, 2019*). World sales are estimated to be 27,200 million US dollars with the employment of

Corresponding authors
Víctor M. Loyola-Vargas,
vmloyola@cicy.mx
Elsa Gongora-Castillo,
elsa.gongora@cicy.mx

approximately 100 million people for cultivation and management (*Bunn et al., 2015*; *ICO, 2019*). Most of the world's production (168 million bags of 60 kg coffee beans in 2018; *ICO, 2019*) is located in small rural areas; this represents a source of income and family business for millions of people (*Martins et al., 2017*; *ICO, 2019*).

The Coffea genus is composed of more than 100 species of perennial woody trees (*Davis et al., 2006*), of which *C. arabica* and *C. canephora* predominate in the world coffee trade, with 63 and 37% of the production, respectively (*ICO, 2019*).

To cover the need for selection programs and market demands, massive propagation techniques such as somatic embryogenesis (SE) have made it possible to obtain a large number of plants of different species, including coffee (*Etienne et al., 2013*; *Loyola-Vargas et al., 2016*). Due to the global importance of the coffee crop, SE has been used for productive and commercial purposes in Central America since 2006 and in Mexico since 2012. However, despite the advantages of using SE as a tool for massive plant propagation, it is still necessary to optimize the scale-up of the process (*Etienne et al., 2013*). Therefore, in-depth knowledge of the genetic and molecular mechanisms that control this process could provide useful information to optimize the SE process. It is of particular interest to understand the changes in the genetic program that allow a somatic cell to become an embryo.

SE is a process that can occur both in nature (Kalanchoe genus) (*Garcês et al., 2007*) and has been translated into laboratories under controlled conditions (*Loyola-Vargas & Ochoa-Alejo, 2016*). Briefly, the SE process consists in cultivating somatic cells under the right conditions to give rise to embryogenic cells. These go through a morpho-physiological process that produces somatic embryos and later complete plants. Unlike zygotic embryogenesis, where the embryo is inside the seed, making it difficult to study, SE allows easy manipulation and control of the culture conditions for the study of morpho-physiological, biochemical, and molecular processes that occur during the development in higher plants. The first stage of SE induction has received particular attention, since knowledge of the key steps that change the genetic program of a cell to become an embryo would serve to improve the biotechnological systems of plant regeneration (*Wójcikowska & Gaj, 2017*).

One of the most important factors during SE induction in different species is the plant growth regulators (PGR) treatment. The use of PGR has been reported in approximately 80% of the protocols for SE induction (*Nic-Can & Loyola-Vargas, 2016*). Most of the PGRs used are auxins, alone or in combination with other regulators, and it is known that there are auxin-related mechanisms functioning through all of the stages of SE induction (*Wójcikowska & Gaj, 2017*). Different components are involved in the mechanism of regulation of the auxins' response genes (*Weijers & Wagner, 2016*; *Sghaier et al., 2018*), in which the key proteins are Transport Inhibitor Resistant 1/Auxin signaling F-Box (TIR1/AFB), Auxin/Indole-3-acetic acid (Aux/IAA) transcriptional co-regulators, and Auxin Response Factors (ARF) binding proteins (*Weijers & Wagner, 2016*).

At low auxin levels, the Aux/IAA proteins form dimers with the ARFs to inhibit ARF activity by binding with the TPL co-repressor (TOPLESS), which results in the repression of the auxin-responsive genes. In contrast, at high auxin levels, Aux/IAA binds to the

SCFTIR1/AFB complex and, as a result, this complex is ubiquitinated and degraded by the 26S proteasome. Thus, ARF proteins are important for the regulation of the responsive-to-auxin genes during transcription (*Li et al., 2016*). Integrating the different layers of knowledge related to the regulation of the Aux/IAA system mediated by ARFs is a key priority for a better understanding of cell development in plants (*Li et al., 2016*).

The arrival of new generation sequencing technologies (NGS) has allowed the massive sequencing of transcriptomic data, leading to the characterization of important genomic resources with high throughput, sensitivity, accuracy, and low cost (*Zhang et al., 2019*). RNA-sequencing (RNA-seq) technology has allowed the sequencing of practically any type of tissue, even uncharacterized biological systems (*Ahn et al., 2014*), including numerous plant species (Arabidopsis, cotton, oil palm) to study development, senescence, growth, responses to different types of stress, and zygotic and somatic embryogenesis (*Chu et al., 2017*; *Góngora-Castillo et al., 2018*). One of the main advantages in using RNA-seq is to identify changes in the genes' expression level under a given condition; for instance, measuring the transcript abundance that controls growth and development of an organism (*Rhee, Dickerson & Xu, 2006*).

Several transcriptome studies have been carried out in Coffea species using RNA-seq technology. As a result of these studies it has been possible to identify potential genes related to agronomic traits; as well as unraveling the genetic mechanisms that operate in different processes of the Coffea plant's development, through different explants like leaves, flowers, fruits (*Ivamoto et al., 2017*; *Yuyama et al., 2016*) and beans (*Cheng, Furtado & Henry, 2017*).

Although little is known about the molecular mechanism that controls SE, NGS and RNA-seq have broadened the scope for studying the first stages of the SE process. Using the RNA-seq approach, it has been possible to identify a set of genes involved in the embryogenic response in species such as cotton (*Cheng et al., 2016*; *Cao et al., 2017*), conifers (*Yakovlev et al., 2016*), papaya (*Jamaluddin, Mohd Noor & Goh, 2017*), wheat (*Chu et al., 2017*), and banana (*Enríquez-Valencia et al., 2019*), among others like Arabidopsis, oil palm, soybean, carrot, grape, alfalfa, and maize (*Cetz-Chel & Loyola-Vargas, 2016*; *Shi et al., 2016*; *Tao et al., 2016*; *Góngora-Castillo et al., 2018*). Thus, the aim of this study is to provide a better understanding of the role of auxins during the induction of SE by identifying differentially expressed genes of *Coffea canephora* when analyzing different stages of SE induction. In particular, it is our intention to understand the changes in the expression profile of ARF and Aux/IAA genes, which can be associated to the changes observed in the phenotype during SE.

## MATERIAL AND METHODS

### Biological material and induction of somatic embryogenesis

The SE induction methodology was carried out according to *Quiroz-Figueroa et al. (2006)*. Briefly, the plantlets were subcultured every four weeks in the maintenance medium without PGR and incubated under photoperiod conditions (150 $\mu$mol m$^{-2}$ s$^{-1}$) 16 h light/8 h dark at 25 $\pm$ 2 °C. To initiate the induction process, the seedlings were previously

incubated for 14 days in the pre-conditioning medium [MS salts (Phyto Technology Laboratories, M524), supplemented with 29.6 μM thiamin-HCl (Sigma, T3902), 550 μM myo-inositol (Sigma, I5125), 0.15 μM cysteine (Sigma, C8277), 16.24 μM nicotinic acid (Sigma, N4126), 9.72 μM pyridoxine-HCl (Sigma, P9755), 87.64 mM sucrose (Sigma, S539) and 0.285% (w/v) Gellan gum (PhytoTechnology Laboratories, G434), combined with 0.54 μM naphthaleneacetic acid (NAA; Sigma, N1145) and 2.32 μM kinetin (KIN; Sigma, K0753) adjusted to pH 5.8], and incubated under the same conditions mentioned above. After 14 days of pre-conditioning, leaves of the second and third pairs were selected in a basipetal direction, and circular explants of eight mm in diameter were cut with the help of a sterile punch. Five explants per 250 mL flask were placed with 50 mL of liquid induction culture medium [Yasuda's medium salts (1985)] supplemented with 5 μM benzyladenine (BA; PhytoTechnology Laboratories, B800) at pH 5.8]. The flasks were incubated in the dark at $25 \pm 2$ °C and shaking (60 rpm) for 56 days.

## Tissue sampling, RNA extraction and sequencing

For RNA extraction and subsequent transcriptome sequencing, 100 mg tissue from *C. canephora* were sampled: (i) before SE induction (pre-embryogenic treatment) at 14 days before the induction (DBI), 9 and 0 DBI; and (ii) under embryogenic conditions at 1 day after induction (DAI), 2 and 21 DAI. Approximately 12 explants of 0.78 cm$^2$ for each time-point (14, 9, 0 DBI and 1 and 2 DAI) and 100 mg of pro-embryogenic mass (21 DAI) were pooled together to obtain total RNA. For the total RNA extraction, Isolate II RNA Plant Kit (Bioline) and treatment of the sample with DNase I, according to the manufacturer's instructions, were used. The quality of total RNA was verified on agarose gel at 1.5% and a Nanodrop (Thermo Fischer Scientific) was used to quantify the RNA. Applied Biological Materials Inc. services were used for RNA-seq library preparation and sequencing (https://www.abmgood.com). Briefly, the integrity of total RNA was assessed using the Agilent 2100 Bioanalyzer with RNA 6000 pico kit (Agilent) before the library construction. RNA samples with an RNA integrity number $\geq 7.5$ were subjected to polyA enrichment followed by first and second strand synthesis and PCR amplification.

## Sequencing and Bioinformatics analysis

A total of 12 single-end libraries were sequenced via Illumina® NextSeq™ 500 system to generate millions of raw reads with a length $\geq$200 bp. Two technical sequencing replicates for each biological sample were conducted corresponding to 14, 9 and 0 days before induction, and 1, 2 and 21 days after induction. The reads' quality was corroborated using FastQC (v.0.11.5) (http://www.bioinformatics.babraham.ac.uk/ projects/fastqc/). The sequences were pre-processed to remove Illumina adapters and low quality reads (Q < 20) using Cutadapt (v. 1.14) (*Martin, 2011*) and FASTQ Quality Trimmer (Part of FASTX Toolkit 0.0.14) (http://hannonlab.cshl.edu/fastx_toolkit/), respectively. Filtered sequences were mapped to the reference genome of *C. canephora* (v1.0) available at http://coffee-genome.org (*Denoeud et al., 2014*; *Dereeper et al., 2015*) using Bowtie2 (v. 2.3.2) (*Langmead & Salzberg, 2012*). Htseq-count (v. 0.10.0) (*Anders, Pyl & Huber, 2015*) was used with default options to quantify the gene expression. Expression counts were

normalized by quantile normalization method and transformed by $\log_2$. To corroborate replicates' similarity, Pearson's correlation coefficient was calculated using RStudio package (v.1.1.456) (*RStudio Team, 2018*; Fig. S1). Differential expression analyses were performed using DESeq2 R-package (v.1.22.1) with a $P < 0.05$ and LFC $\leq 1$ or LFC $\geq 1$ (*Love, Huber & Anders, 2014*). The heatmaps were generated using the ggplot2 package for R (*Wickham, 2016*). The InteractiveVenn tool was used to create the Venn diagram (*Heberle et al., 2015*). The orthology analysis was performed using the program OrthoFinder (v.2.2.7) (http://www.stevekellylab.com/software/orthofinder), comparing the amino acid sequences of ARF and Aux/IAA proteins between *C. canephora* and *Arabidopsis thaliana*.

### Quantification of relative expression by qPCR

Total RNA samples corresponding to days 14, 9 and 0 DBI, and 1 and 21 DAI were converted to cDNA with the RevertAid H Minus First Strand cDNA Synthesis kit (Thermo Scientific, K1632) and quantified in a Nanodrop 2000 (Thermo Scientific). The primers used are listed in Table S1. Real-time qPCR quantification was performed with the Express Sybr GreenER qPCR Supermix Universal kit (Invitrogen, A10314), on a StepOne Real-Time PCR System (Applied Biosystems) with three replicates per sampling day, using the cyclophilin gene as the internal reference (*Goulao, Fortunato & Ramalho, 2012*). The relative expression data were obtained by the method $2^{-\Delta\Delta CT}$ (*Livak & Schmittgen, 2001*).

## RESULTS

### Induction of somatic embryogenesis

SE induction was carried out in leaf explants of *C. canephora* plantlets cultivated *in vitro*. In Fig. 1 we describe how our SE process is divided into two stages, pre-conditioning and induction, and the characteristics of each one. Plantlets were placed in MS medium supplemented with NAA and KIN for a period of 14 days for pre-conditioning (before induction) (Fig. 2A). At the end of this stage, the second and third pairs of leaves were used to obtain circular explants. To start the SE induction, the explants were placed in Yasuda's medium added with BA. With a naked eye, no changes were observed in the explants during the first 72 h of induction. However, apro-embryogenic mass around the explants was observed after 14 days (Figs. 2B, 2C). After 56 days in the induction medium, each one of the explants showed approximately 300 somatic embryos at different stages of development (globular, heart, torpedo, cotyledonar) located at the periphery and protruding into the culture medium (Fig. 2D). To promote germination, the somatic embryos obtained were transferred to MS semi-solid culture medium free of PGR (Fig. 2E). All of the embryos continued their development until they achieved complete seedlings (Fig. 2F).

### Analysis of gene expression during the induction of somatic embryogenesis of *C. canephora*

Samples of *C. canephora* leaf explants before SE induction (14, 9 and 0 DBI) and under embryogenic conditions (1, 2 and 21 DAI) were collected for RNA sequencing and analysis. Twelve cDNA libraries were constructed and sequenced with the Illumina HiSeq

| Stage | Pre-conditioning | | | Induction | | |
|---|---|---|---|---|---|---|
| Days | 14DBI | 9DBI | 0DBI | 1DAI | 2DAI | 21DAI |
| Medium | MS | MS + 0.54 µM NAA 2.32 µM KIN | MS + 0.54 µM NAA 2.32 µM KIN | Yasuda + 5 µM BA | Yasuda + 5 µM BA | Yasuda + 5 µM BA |
| Tissue | | | | | | |

**Figure 1** **Somatic embryogenesis induction in *C. canephora*.** Description of the stages, days, culture medium and tissues that comprise the process of somatic embryogenesis induction in *C. canephora*.

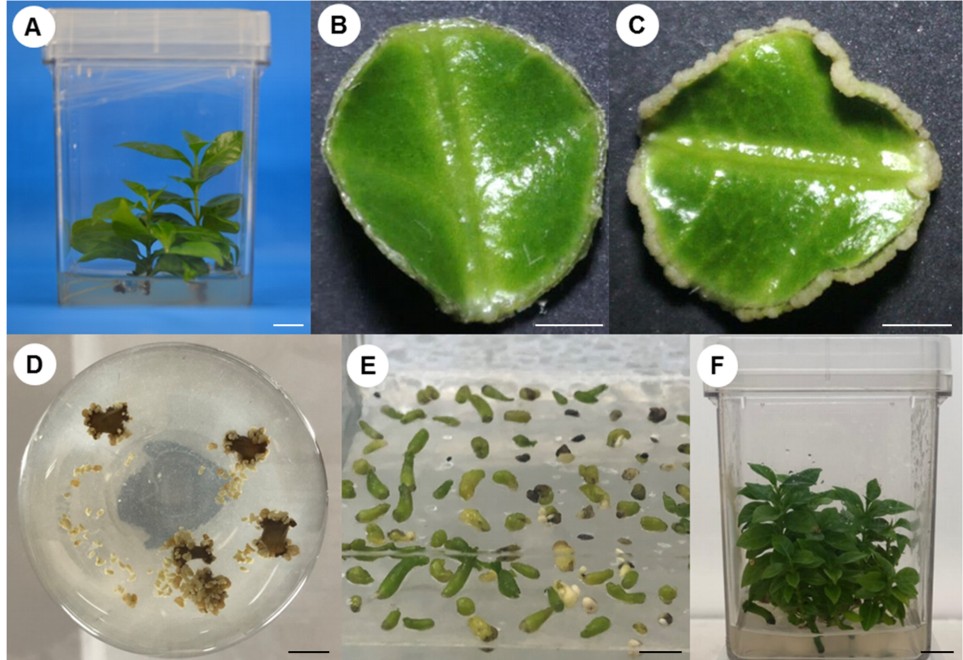

**Figure 2** **Process of somatic embryogenesis induction in *C. canephora*.** (A) Plantlet in pre-conditioning stage in MS medium added with 0.54 µM NAA and 2.32 µM KIN. (B) Explant at 14 days after induction in Yasuda's medium (*Yasuda, Fujii & Yamaguchi, 1985*) supplemented with 5 µM BA. (C) Explant 21 days after induction of SE. (D) Explants 56 days after induction of SE. (E) Embryos at germination stage, in MS medium without PGR. (J) Complete plantlets regenerated from somatic embryos. Scale bars: one cm (A, D, F), 0.2 cm (B, C), 0.5 cm (E).

500 platform and a total of 534,469,789 reads were obtained and pre-processed for quality. The rate of recovered sequences ranged from 88 to almost 100% (Table 1). High-quality sequences were mapped to the *C. canephora* genome (*Denoeud et al., 2014*; *Dereeper et al., 2015*). The results showed an overall alignment rate that ranged from 54 to 77% (Table 1; Table S2). The expression abundances were calculated for each transcriptome at different

**Table 1 Sample description and basic statistics of RNA-seq data libraries of *C. canephora* transcriptome.**

| Sample | Treatment | Raw sequences | High-quality sequences | Rate of recovered sequences (%) | Overall alignment rate (%) |
|--------|-----------|---------------|------------------------|--------------------------------|----------------------------|
| 14_1 DBI | Pre-conditioning | 42,157,360 | 38,064,798 | 90.29 | 60.97 |
| 14_2 DBI | | 44,738,607 | 44,737,035 | 100.00 | 75.17 |
| 9_1 DBI | Pre-conditioning | 43,459,278 | 40,731,459 | 93.72 | 66.03 |
| 9_2 DBI | | 43,628,937 | 42,369,506 | 97.11 | 68.52 |
| 0_1 DBI | Pre-conditioning | 40,924,936 | 38,247,628 | 93.46 | 69.29 |
| 0_2 DBI | | 49,032,211 | 49,012,743 | 99.96 | 73.88 |
| 1_1 DAI | Induction | 38,121,519 | 33,791,596 | 88.64 | 66.58 |
| 1_2 DAI | | 49,094,305 | 44,718,468 | 91.09 | 70.24 |
| 2_1 DAI | Induction | 62,157,781 | 54,777,604 | 88.13 | 66.43 |
| 2_2 DAI | | 61,989,925 | 61,987,800 | 100.00 | 76.20 |
| 21_1 DAI | Induction | 39,436,806 | 34,526,731 | 87.55 | 54.13 |
| 21_2 DAI | | 51,508,818 | 51,504,421 | 99.99 | 76.95 |

time points and biological replicates were analyzed by Pearson's correlation coefficient. The lowest correlation (0.77) was only obtained for the replicates of day 9 DBI; this may be due to technical problems during sequencing, or due to variation in transcript expression (Fig. S1).

As expected, the analysis of abundance expression for each transcriptome showed that important changes occur in the gene expression profiles when the explants are transferred to Yasuda's medium (1 DAI) (Fig. 3). Interestingly, the analysis of the expression profiles revealed two well-defined clades based on expression counts. The "yellow" clade shows highly expressed genes in which a set of genes are expressed during the entire process of SE. The "blue" clade shows medium-to-low expressed genes, including off genes. The number of minimally expressed and switched off genes is significantly larger than the number of genes in the "yellow" clade, suggesting that many biological functions might not be involved in the SE process (Fig. 3).

## Expression profiles of ARF and Aux/IAA gene family in *C. canephora* genome

The analysis of the *C. canephora* genome revealed a total of 22 ARF genes. Of these, 17 were expressed throughout the entire SE process (Fig. 4A). Additionally, 14 Aux/IAA genes were identified and all of them were expressed in at least one point of the SE process (Fig. 4B). An orthology analysis with *A. thaliana* showed that ARF5, ARF6 and ARF9 genes of *C. canephora* are orthologous to the same ARF in *A. thaliana*, whereas ARF18 (Cc06_g03950) in *C. canephora* is ortholog to ARF16 in *A. thaliana*. Similarly, Aux/IAA7 of *C. canephora* is ortholog to Aux/IAA7 of *A. thaliana*. However, identification of the Aux/IAA12 gene ortholog was not possible (Table S3).

The expression profile analysis of ARF genes revealed that the ARF5 gene is minimally expressed in the majority of the stages except in 21 DAI, during which it increases its expression level (Fig. 4A). There is a correlation between the expression changes of this

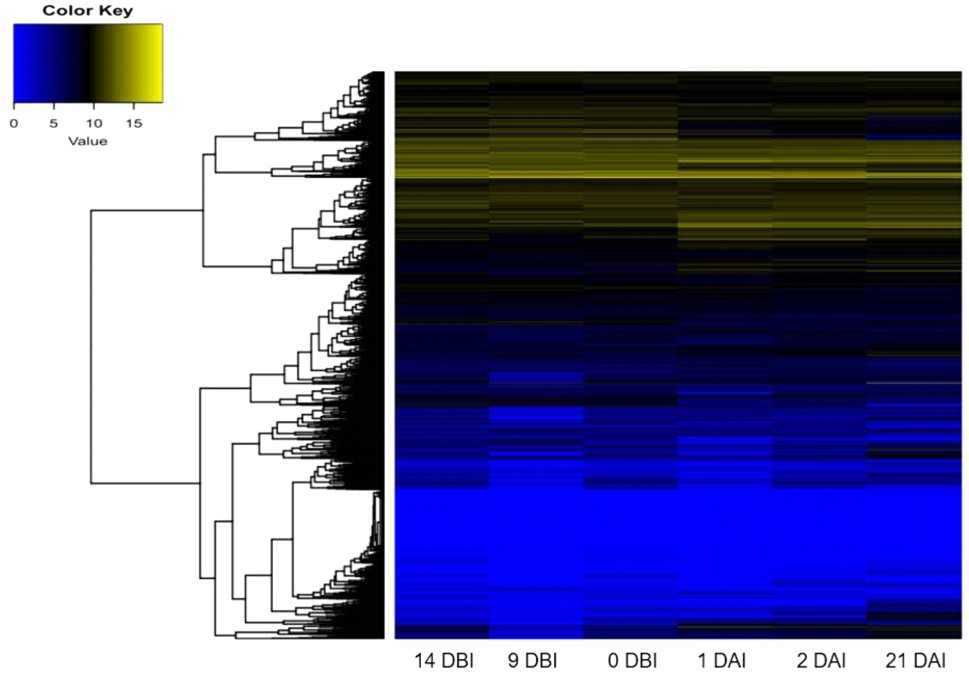

**Figure 3 Expression profile of genes of *C. canephora*.** Heatmap of the expression profile of 25,574 genes of *C. canephora* during the different stages of the process of SE induction.

gene and the appearance of embryonic structures as observed in Fig. 2C. The ARF6 gene is highly expressed in all six stages; however, it reaches its maximum at 2 DAI. Conversely, the expression level of the ARF9 gene oscillates over time. The expression of this gene decreases during the pre-conditioning stage (14, 9, 0 DBI) and then slightly increases in the early stages of induction (1, 2 DAI). In the case of the ARF18 gene, its expression decreases considerably from 1 DAI on. The levels of expression of the gene Aux/IAA12 remain stable without notable changes throughout the process, as well as the gene Aux/IAA7, except for 21 DAI, when its expression decreases (Fig. 4B).

A differential gene expression analysis was performed using the 14 DBI sample as reference control. We used 14 DBI as a control since this is the tissue at the beginning of the experiment, where plantlets are maintained on MS medium without PGR (Fig. 1). Thus, the following comparisons were analyzed to obtain the differentially expressed genes among them (DEG): 14 DBI vs. 9 DBI; 14 DBI vs. 0 DBI; 14 DBI vs. 1 DAI; 14 DBI vs. 2 DAI; and 14 DBI vs. 21 DAI (Fig. 5). The analysis of the pre-conditioning stage (14 DBI vs. 9 DBI, and 14 DBI vs. 0 DBI) revealed 557 and 26 DEG, respectively. Comparisons for the first days of induction (14 DBI vs. 1 DAI, and 14 DBI vs. 2 DAI) showed an increased number of DEG: 4,570 and 3,286, respectively; and 5,319 DEG were observed for late time of SE induction (14 DBI vs. 21 DAI) (Fig. 5). Up- and down-regulated genes were identified for each comparison. The results showed that approximately 75%, 55% and 56% of genes were down-regulated when the control (14 DBI) was compared against 9 DBI, 1 DAI and 2 DAI, while comparing 14 DBI vs. 21 DAI showed a similar proportion of

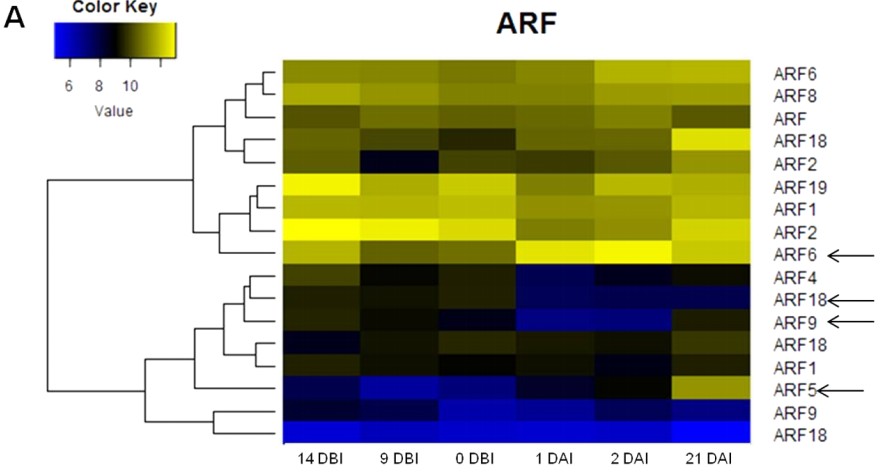

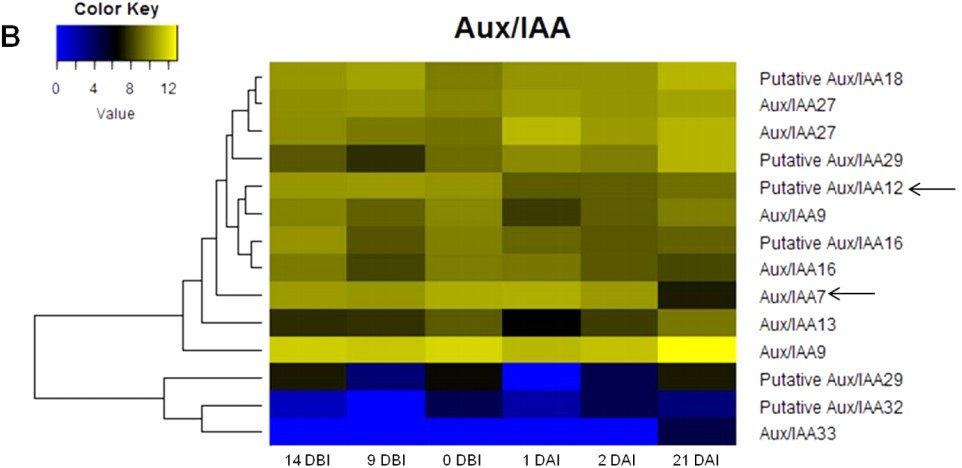

**Figure 4** **Expression profile of genes involved in IAA signaling during the different stages of the process of SE induction.** (A) Auxin Response Factor (ARF). (B) Aux/IAA. (Arrows indicate the genes selected for quantification by qPCR).

up- and down-regulated genes, with 51% and 49%, respectively (Fig. 5). These results are in line with the observations in Fig. 3, in which the transcriptome's profile showed that majority of the genes are down-regulated. Also in Fig. 3, we can see that the expression pattern of 0 DBI is more similar than the rest of the days to that in 14 DBI; this result was confirmed by the DEG analysis at this time (14 DBI vs. 0 DAI). These results may suggest that the most dramatic changes in gene expression occur during the first days of passing the explants to a new culture media with different composition.

A Venn diagram showing the interaction between the DEGs of the different comparison sets was created (Fig. 6). Interestingly, only five genes remain differentially expressed consistently throughout the whole ES process; however, most of them are down-regulated. These five genes were identified according to the functional annotation as protein involved in salt tolerance SIS (Salt-Induced Serine-rich) (Cc06_g02060), adenine nucleotide alpha hydrolases-like superfamily protein (Cc05_g05700), NAC domain-containing protein

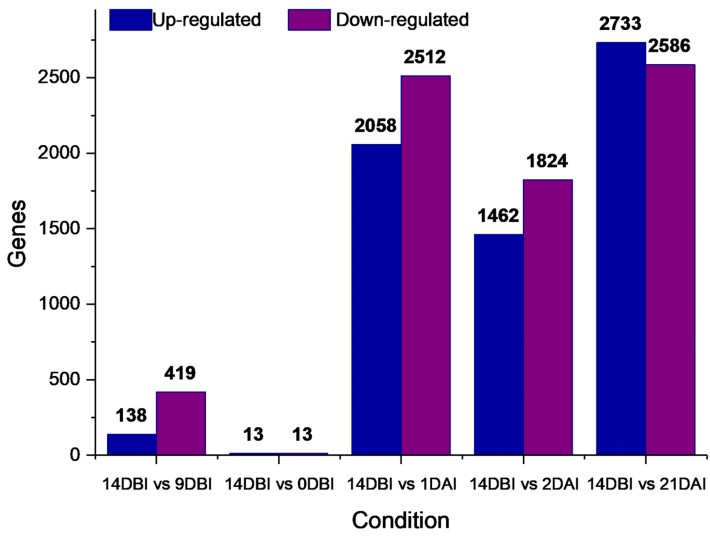

**Figure 5** **Differentially expressed genes.** Number of differentially expressed genes at different stages of the process of SE induction, when comparing 14 DBI (control) against 9 and 0 DBI, and 1, 2 and 21 DAI.

(Cc02_g33930), mitogen-activated protein kinase kinase kinase (Cc07_g06080) and a hypothetical protein (Cc00_g04350).

Stage-specific genes were identified for each comparison and results revealed that 1 DAI and 21 DAI stages showed the bulk of these genes (1,602 and 2,511, respectively) (Fig. 6). Of these, 50.7% and 64.3% are up-regulated, and consequently, 49.3% and 35.7% are down-regulated at 1 DAI and 21 DAI, respectively. These results are consistent with the observed phenotypic variation, since at the 1 DAI stage the explants are changed to Yasuda's induction medium, and at 21 DAI stage embryos are observed surrounding the leaf explant (Fig. 2C). Comparisons at 9 DBI and 2 DAI showed 196 and 289 stage-specific genes, respectively. Of these, 19.9% were up-regulated and 80.1% were down-regulated when comparing the control to 9 DBI; and 39.4% were up-regulated and 60.6% were down-regulated at the control vs. 2 DAI comparison. Additionally, only one gene was distinguished when comparing the control to 0 DBI. This gene was identified as an ABC transporter according to functional annotation, and it is up-regulated at this pre-conditioning stage (Fig. 6).

The expression analysis revealed that nine ARFs genes were differentially expressed in three different stages of the SE process. ARF19, ARF4, ARF9, and ARF18 (Cc06_g03950) genes were down-regulated, while ARF6, ARF5 and ARF18 (Cc01_g11020) genes were up-regulated. Likewise, seven different Aux/IAAs genes were differentially expressed. IAA9, IAA12, IAA29, and IAA7 were down-regulated, while IAA13, IAA33 and IAA29 genes were up-regulated (Table 2).

## Quantification of the level of relative expression

The ARF18 (Cc06_g03950), ARF9 (Cc08_g16330), ARF6 (Cc09_g08740) and ARF5 (Cc10_g01900) genes, as well as the Aux/IAA7 (Cc03_g04670) and Aux/IAA12 (Cc01_g17790) genes, were selected as candidate genes to measure expression by qPCR

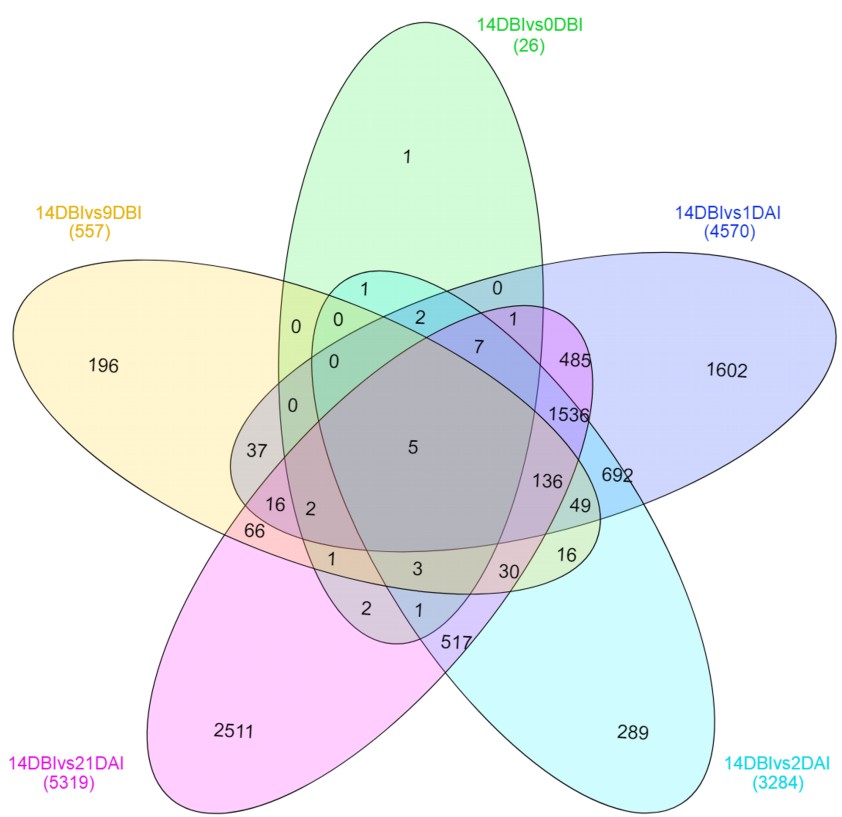

**Figure 6** **Venn diagram of differentially expressed genes between each comparison.** The overlapping regions correspond to the number of DEG shared between each point of the process of SE induction.

(Table S1). The qPCR results showed that most of the ARF genes were less expressed at the pre-conditioning stage (14, 9 and 0 DBI) compared to the last day of the induction stage (21 DAI); except for the ARF6 gene, which was highly expressed only at 1 DAI stage, during which the explants are changed to Yasuda's induction medium, suggesting that this gene might play a key role at the first stages of SE induction. As expected, the ARF5, ARF9 and ARF18 (Cc06_g03950) genes increased until 21 DAI, while the expression of the ARF6 gene was suppressed entirely (Fig. 7).

Conversely, the Aux/IAA7 gene was highly expressed at the pre-conditioning stage and its expression level decreased at the induction stage. The Aux/IAA12 gene expression slightly increased at 9 DBI but then decreased at the induction stage. The results suggest that the Aux/IAA genes play a key role during the pre-conditioning stage of the SE process (Fig. 7). Most of the results for qPCR analysis (Fig. 7) are similar to those found in the expression profile of the transcriptome (Fig. 4), except for the ARF18 gene, where the expression profile is not the same in both analyses. We suggest that this difference between the two analyses is due to a biological variation between the different samples used for each one.

Table 2  Fold change values of differentially expressed ARF and Aux/IAA genes during the process of SE induction in *C. canephora*.

| Gene ID | Contig name | 14 DBI vs. 1 DAI | | | 14 DBI vs. 2 DAI | | | 14 DBI vs. 21 DAI | | |
|---|---|---|---|---|---|---|---|---|---|---|
| | | LFC | *P* value | *P*-adj | LFC | *P* value | *P*-adj | LFC | *P* value | *P*-adj |
| ARF2 | Cc00_g12260 | −2.1 | 2.5E–10 | 1.0E–08 | −1.8 | 6.7E–06 | 1.3E–04 | – | – | – |
| ARF4 | Cc01_g11410 | −2.6 | 1.6E–04 | 1.5E–03 | – | – | – | – | – | – |
| ARF5 | Cc10_g01900 | – | – | – | 1.3 | 4.7E–03 | 3.2E–02 | 3.4 | 2.7E–11 | 8.2E–10 |
| ARF6 | Cc09_g08740 | – | – | – | 1.1 | 7.8E–03 | 4.8E–02 | – | – | – |
| ARF9 | Cc08_g16330 | −2.7 | 9.5E–07 | 1.7E–05 | −2.5 | 3.7E–06 | 7.4E–05 | – | – | – |
| ARF18 | Cc01_g11020 | – | – | – | – | – | – | 2.0 | 1.8E–03 | 9.8E–03 |
| ARF18 | Cc06_g03950 | −1.9 | 1.2E–04 | 1.1E–03 | −1.7 | 3.7E–04 | 4.0E–03 | −1.8 | 1.7E–03 | 9.4E–03 |
| ARF19 | Cc00_g00210 | −2.0 | 1.9E–04 | 1.7E–03 | – | – | – | – | – | – |
| IAA7 | Cc03_g04670 | – | – | – | – | – | – | −3.6 | 1.2E–07 | 1.9E–06 |
| IAA9 | Cc07_g07780 | −2.3 | 2.1E–03 | 1.3E–02 | – | – | – | – | – | – |
| *P. IAA12 | Cc01_g17790 | −1.6 | 7.5E–04 | 5.5E–03 | −1.5 | 2.5E–03 | 1.9E–02 | – | – | – |
| IAA13 | Cc03_g06400 | – | – | – | – | – | – | 1.8 | 5.2E–04 | 3.5E–03 |
| *P. IAA29 | Cc06_g08150 | −11.1 | 2.1E–03 | 1.3E–02 | −4.2 | 4.6E–04 | 4.8E–03 | – | – | – |
| *P. IAA29 | Cc08_g00560 | – | – | – | – | – | – | 2.0 | 5.5E–03 | 2.6E–02 |
| IAA33 | Cc06_g13230 | – | – | – | – | – | – | 7.3 | 1.0E–03 | 6.1E–03 |

**Notes.**
No differentially expressed genes were found at 9 DBI and 0 DBI. Dashes means that genes were not detected as differentially expressed. *P, putative.

# DISCUSSION

With the release of the *C. canephora* genome (*Denoeud et al., 2014*), a better understanding of this important crop is made possible. It has certainly expanded the number of questions that can be asked; for instance, what molecular mechanisms are responsible for the changes in a somatic cell that allow it to give rise to an embryogenic cell? Even though the knowledge related to changes in the genetic program of plant cells has grown exponentially in recent years, there are still several aspects of the process that remain unknown (*Fehér, Bernula & Gémes, 2016*; *Cetz-Chel & Loyola-Vargas, 2016*). Particularly, it is not well understood if the initiation of SE responds to the same set of signals in all the species (*Loyola-Vargas et al., 2016*; *Loyola-Vargas & Ochoa-Alejo, 2016*; *Loyola-Vargas & Ochoa-Alejo, 2016*; *Sisodia & Bhatla, 2018*).

The RNA-Seq technology has allowed us to have a complete picture of the genes expressed during SE induction. Transcriptomic studies have showed that the most important and determining changes begin at the start of the SE induction stage (1 DAI) and at the beginning of the development of the first structures (21 DAI) or, in other words, at the moment of the shift from a non-embryogenic state to an embryogenic one (*Cheng et al., 2016*; *Shi et al., 2016*; *Tao et al., 2016*; *Cao et al., 2017*; *Chu et al., 2017*; *Enríquez-Valencia et al., 2019*). *Cao et al. (2017)* identified the main regulation phase of SE initiation in cotton to occur between three hours and three days after induction. As is the case in our results, most of the dramatic changes are visualized during the transition from the pre-conditioning stage to the first hours of the induction stage and the 21 DAI where embryogenic structures can be observed.

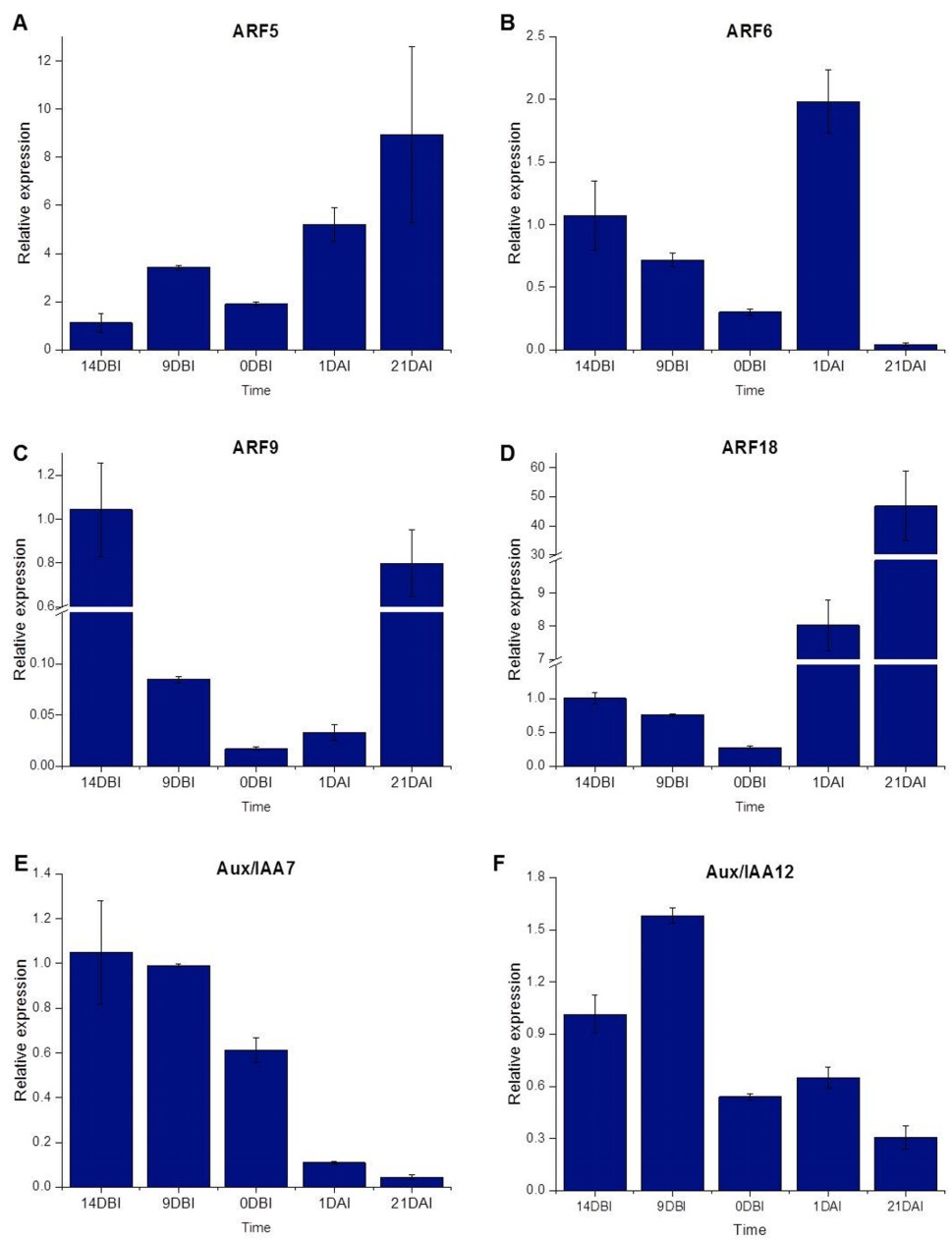

**Figure 7  Relative expression level.** Quantification of the relative expression level (qPCR) of certain genes related to auxin signaling during SE induction in *C. canephora*.

The ARF and Aux/IAA genes play a primordial role in the perception and signaling of auxins and the consequent triggering of cellular responses. However, the importance of the ARFs and Aux/IAA genes in several processes of plant development are still not systematically characterized (*Li et al., 2015*).

Both ARFs and Aux/IAA proteins work as transcriptional regulators. While the Aux/IAA repress the auxin response genes, ARF proteins activate or repress the transcription

depending on the middle region structure of the protein (*Liscum & Reed, 2002*). Aux/IAA encode short-lived nuclear proteins that act as repressors of auxin-mediated transcriptional activation and, although not all Aux/IAA respond to auxins, those that do differ in their sensitivity and type of activity response to this PGR (*Liscum & Reed, 2002*). In this way, the Aux/IAA proteins control the expression by means of the ARF protein activity to which they join with different levels of specificity. However, the expression of a given set of ARF and Aux/IAA genes vary from one species to another, and even from one tissue type to another (*Jain & Khurana, 2009*; *Rademacher et al., 2011*).

Several studies have allowed the identification of the role of some of the 23 ARFs and 29 Aux/IAAs in the presence of auxins at different stages of plant development, using mutants of loss or gain of function in *A. thaliana* (*Liscum & Reed, 2002*). To mention a few, ARF5 is critical for the formation of the embryonary axis and the embryogenic root, and the development of the flowers, as well as part of the vascular tissue; ARF6 acts in the maturation of the flowers and, in tomato, ARF9 regulates cell division (*Li et al., 2015*; *Li et al., 2016*; *Weijers & Wagner, 2016*; *Chen et al., 2017*). Regarding Aux/IAA proteins, studies in *A. thaliana* have shown that mutations in the Aux/IAA7 gene produce short hypocotyl phenotypes and deformities in the leaves; mutations in the Aux/IAA12 gene cause root abnormalities (*Liscum & Reed, 2002*).

The ARFs and Aux/IAAs interactions are not well known, but the most studied case is the interaction of the MONOPTEROS (MP/ARF5) and BODENLOS (BDL/IAA12) genes. They are essential effectors of the action of auxins in the embryo. The mutants produce defects in the embryogenic pattern, in particular in the formation of the embryonic axis, the formation of the cotyledons and the apical and radicular meristems (*Weijers & Wagner, 2016*). The BDL/IAA12 protein usually is degraded in response to auxins; therefore, a mutation in BDL/IAA12 prevents its degradation, causing abnormal embryogenic phenotypes. Until now, the mechanisms of regulation of auxins by other interactions of ARFs and Aux/IAAs directly related to embryogenesis have been unknown (*Weijers & Wagner, 2016*; *Mironova et al., 2017*).

In our results, we observed a decrease in the expression of Aux/IAA7 and Aux/IAA12 as ARF5, ARF9 and ARF18 (Cc06_g03950) increased (Fig. 7). *Mironova et al. (2017)* mention that the degradation of Aux/IAA12 associated with the signaling of ARF5 is necessary for auxin signals to be translated into programs of morphogenesis and cell development. Also, it has been determined that MP/ARF5 can self-regulate its transcription and that of BDL/IAA12, as well as that of other Aux/IAA genes, through the action of auxins. In this way, it is inferred that the auxins function as a trigger causing the degradation of BDL/IAA12 (*Chandler, 2016*).

*Wójcikowska & Gaj (2017)* observed that the ARFs that are most expressed during SE in Arabidopsis were ARF5, ARF6, ARF8, ARF10, ARF16 and ARF17, with ARF5 and ARF10 having the highest accumulation of transcripts; while the least expressed were ARF1, ARF2, ARF3, ARF11, and ARF18. Similarly, our differential expression results showed that the highest levels of expression are presented by the ARF5 and ARF6 genes (Fig. 4A), as well as by the fact that the ARF2 and ARF18 (Cc06_g03950) are down-regulated (Table 2). However, in the qPCR test, ARF18 (Cc06_g03950) showed the highest relative expression
levels, followed by ARF5. Another similarity found was that ARF18 (Cc06_g03950) had a reduction in its activity in the early stages of the induction of SE, similar to the results reported here, although for the last day of the induction process it increased its levels of expression. On the other hand, the ARF5 gene also showed higher expression and the level of transcripts increased in the early stages of SE. This observation suggests that there is a close relationship between this gene and the embryogenic transition of the cells.

Several studies have demonstrated that ARF6 usually is co-expressed with ARF8. In *Dimocarpus longan*, an increase of ARF6 is necessary to initiate the development of somatic embryos, while ARF8 is related to the transition to globular and cotiledonar stages in embryos (*Lin et al., 2015*). In *arf6/arf8* Arabidopsis double mutants it was found that ARF6, together with ARF8, plays an important role during SE induction by mediating auxin signaling and leading to the activation of related genes. Also, the production of jasmonic acid decreases, which could be related to the blocking of SE (*Su et al., 2016*; *Kumar & Van Staden, 2017*).

In zygotic embryogenesis it is well documented that ARF9 participates in the suspensors cell as well as in the protoderm of the lower tier in the pro-embryo (*Rademacher et al., 2011*). However, in SE, ARF9 was found to play significant roles in the regulation of SE induction from *Lilium oriental* callus by increasing its expression during the first days of culture (*Chen et al., 2019*). This finding is contrary to our results that showed that ARF9 had a low expression during the first days of induction. Nevertheless, in oil palm a reduction of ARF9 during the first days of culture in the presence of 2,4-D was found, but after 7 days this condition changed by increasing the expression, suggesting that the ARF9 had overcome an initial suppression due to the addition of exogenous auxin to the culture medium (*Ooi et al., 2012*). Thus we can suggest a similar situation in our model, since ARF9 expression decreased during the pre-conditioning stage where exogenous auxin was added and rose until the last day of induction.

About the expression of the Aux/IAA genes, *Yang et al. (2012)* observed that most of the transcription of these genes decreased during the transition stage to embryogenesis, although the expression of these genes increased during the development of the embryos. In the same way, our results indicate that Aux/IAA7 and Aux/IAA12 diminish their expression gradually as the induction process of SE progresses, reaching values close to zero.

Additionally, of all of the genes differentially expressed, only five remained constant in that condition during the whole process of SE induction. Of these genes, the four described below are related to biotic and abiotic stress responses, and to processes of vegetal development such as embryogenesis.

SIS proteins (Salt-Induced Serine-rich) and the adenine nucleotide alpha hydrolase-like superfamily play an important role in the protective function of the endosperm on the embryo (together with other factors and proteins) in *Brassica napus* (*Lorenz et al., 2014*).

Proteins with a NAC domain are transcription factors involved in different processes of plant development and have recently received special attention due to their implication in the responses to biotic and abiotic stress, as well as their interaction with certain PGR. They are also essential regulatory proteins in the process of cell proliferation and plant regeneration (*Puranik et al., 2012*). Similar to the observations in this work, a NAC protein

was identified as differentially expressed during the SE of *Citrus sinensis* callus (*Ge et al., 2012*). On the other hand, NAC proteins have been identified as the target gene of BABY BOOM (BBM). Despite the study being carried out in non-embryonic tissues, it is remarkable the importance of BBM genes during cellular development and embryogenesis in plants (*Passarinho et al., 2008*). Also, NAC proteins are essential regulators during plant regeneration as found in Arabidopsis stems, where these proteins were expressed when a wound was made on the stem and auxin accumulated to initiate a proliferation process (*Ikeuchi, Sugimoto & Iwase, 2013*).

For its part, the mitogen-activated protein kinase kinase kinase (MAPKKK) play important roles in the transduction of intra- and extracellular signals related to stress and plant cell development by regulating diverse processes, such as the homeostasis of reactive oxygen species and the response to PGR in Arabidopsis (*Nakagami et al., 2006*), leaf senescence, cell division, and lateral root formation, among others (*Xu & Zhang, 2015*).

Another interesting result found was that, in the comparison 14 DBI vs. 0 DBI, only one gene was uniquely expressed at this condition which was not present in the other comparisons: an ABC transporter. Members of this family are implicated in the transport of auxinic compounds such as IAA, IBA, and even synthetic analogues (*Geisler et al., 2017*). In a previous study of the endogenous levels of auxins, it was found that during the pre-conditioning stage there is an important increase in IAA and IBA (*Ayil-Gutiérrez et al., 2013*), so this may suggest the importance of the transporter in the process.

The results obtained from this research are relevant since the combination of the pairs of Aux/IAAs and ARFs determine their role during development (*Jain & Khurana, 2009*). There are two facts to keep in mind: first, if any ARF could interact with any Aux/IAA protein, there would be more than 600 possible pairwise combinations (*Weijers et al., 2005*); second, not all the Aux/IAAs respond to auxin (*Paponov et al., 2008*). Together, the results presented in this research and those of the literature suggest that the paired Aux/IAAs and ARFs are essential for SE to be carried out.

## CONCLUSION

There are several plant development processes related to auxin signaling mediated by genes from the ARF and Aux/IAA families. The role of each of these genes can vary from one species to another and in the genus Coffea, specifically in the species *C. canephora*, there are no records of their expression during SE. As a first approach, with the use of bioinformatics tools, it was possible to identify ARF and Aux/IAA genes differentially expressed through the entire induction SE process. A family of 22 ARF genes was found in *C. canephora* genome; 17 are expressed in our study model, and eight of them are differentially expressed in different stages of the SE process. A total of 14 Aux/IAA genes were found in the *C. canephora* genome, and although all are expressed in at least one point of the SE process, only seven are differentially expressed. The quantitative analysis by qPCR revealed that the ARF18 (Cc06_g03950) and ARF5 genes are highly expressed at 21 DAI. Conversely, the expression of Aux/IAA7 and Aux/IAA12 genes varies from high to low through the

different SE stages. The results presented in this study provide valuable information for our understanding of the underlying molecular mechanisms by which a somatic cell gives rise to an embryogenic cell.

### Funding

This work was supported by the National Council of Science and Technology (FS-1515 to Víctor M. Loyola-Vargas). The funders had no role in study design, data collection and analysis, decision to publish, or preparation of the manuscript.

### Grant Disclosures

The following grant information was disclosed by the authors:
National Council of Science and Technology: FS-1515.

### Competing Interests

The authors declare there are no competing interests

### Author Contributions

- Ana O. Quintana-Escobar performed the experiments, analyzed the data, prepared figures and/or tables, authored or reviewed drafts of the paper, approved the final draft.
- Geovanny I. Nic-Can performed the experiments, approved the final draft, the somatic embryogenesis and molecular biology experiments.
- Rosa María Galaz Avalos performed the experiments, approved the final draft, the somatic embryogenesis and molecular biology experiments.
- Víctor M. Loyola-Vargas conceived and designed the experiments, analyzed the data, contributed reagents/materials/analysis tools, authored or reviewed drafts of the paper, approved the final draft.
- Elsa Gongora-Castillo conceived and designed the experiments, analyzed the data, authored or reviewed drafts of the paper, approved the final draft.

### DNA Deposition

The following information was supplied regarding the deposition of DNA sequences:
Data is available at NCBI GEO: GSE128888.

### Data Availability

The raw data are available in the Supplemental Tables.

### Supplemental Information

Supplemental information for this article can be found online at http://dx.doi.org/10.7717/peerj.7752#supplemental-information.

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
