# Peer review of "Transcriptome analysis of the induction of somatic embryogenesis in Coffea canephora and the participation of ARF and Aux/IAA genes"

_PeerJ, doi:10.7717/peerj.7752_

## Round 0.1 · original submission · Major Revisions

Three reviewers have evaluated your manuscript and suggest various corrections. Please consider the comments, to improve the clarity of the presentation of your work (English, error bars, references etc.).

[]

Reviewer 1 ·

Basic reporting

The authors of the study analysed expression changes during SE in Coffea canephora by using rna-seq ad qPCR. Generally speaking it is a good study but there are some concerns needing to be addressed.

Experimental design

1. line146 why to choose these days (14, 9, 0 DBI and 1, 2, 21 DAI) for RNA-seq DEG experiments? Besides, why to choose these 5 comparisons must be clearly stated. 14 and 21 likely stand for two and three weeks. What about the others? These experimental designs are essential to conclusions. Why were not DBI0 VS DAI2,14,21 analysed? They look to me to make sense because they are to compare between before induction and after induction. I'd like to know why not choosing these comparisons. Please correct me if I was wrong or overlooked it.
2. In figure 4, can the authors classify why 0DBI showed less DEGs than 9DBI? Because 0DBI stands for the later developmental stage I think 0DBI should have more DEGs compared to 14DBI.
3. DEGs in the study didn't consider the change of fold (line 176-177), which is typically standard in DE analysis. This perhaps leads to the high fraction of DEGs showed in the line241-245. I recommend the authors to carefully revise it. Or if the authors think it's fine not to set an arbitraty cutoff of fold change for DEs I'd like to know the reasons.

Validity of the findings

1. The correlations between replicates for each range from 0.7 to 1.0 (line 174-177, fig. S1). However, 0.7 between replicates is not very high as compared to other studies. Is there any reason for that?
2. Why is there not an error bar shown in the qPCR figure 6? I do not think the authors have stated that there are replicates for qPCR experiments. If so why didn't the authors have any replicates for qpcr or simply because they were done but not indicated in the manuscript?
3. citations are needed to support the discussion at line332

Additional comments

1. Figure 2 what are the values indicating? what is the shaded area in the right part of the figure?
2. what are the values in table 2 representing?
3. in table s2: at 1 times and > 1 times should be changed to uniquely aligned and multiply aligned reads
4. typos:
line121: this, the aim?
line 124-125: rephrase please
line276 lowly expressed
line277: in which it increases
382: showed to have

·

Basic reporting

The work is fairly well-written and sheds light on the involvement , whose findings lead to a better understanding of the genes involved in auxin signaling pathway as well as their expression profiles during the somatic embryogenesis (SE) process in Coffea canephora.

Authors clearly indicate a knowledge gap in the field of study, establishing the context, background and/or importance of the topic. It is established the desirability of the research, and listing the research questions and hypotheses, and explaining the significance and validity of the study.

Experimental design

What is presented is an original primary research pretty much in line with the scope of the Peer J. The research question is well- defined, relevant and meaningful, and the findings do contribute to expand and advance the current knowledge on coffee SE. It is clearly stated how the research fills an identified the knowledge gap of SE in coffee.

As far as the methodology is concerned a rigorous investigation performed to a high technical and ethical standard, with methods described with sufficient detail and information to be replicated elsewhere.

Validity of the findings

Regarding the findings the research group is a leading team in the targeted topic and the data presented here are robust, statistically sound, and well-controlled. However, in Figure 6 there is no indication whatsoever of a statistical analysis to compare expression levels!

Remarkably, the conclusions are appropriately stated, pretty much connected to the original question investigated, and limited to those supported by the results.

Additional comments

The work is fairly well-written and sheds light on the involvement , whose findings lead to a better understanding of the genes involved in auxin signaling pathway as well as their expression profiles during the somatic embryogenesis (SE) process in Coffea canephora.

Authors clearly indicate a knowledge gap in the field of study, establishing the context, background and/or importance of the topic. It is established the desirability of the research, and listing the research questions and hypotheses, and explaining the significance and validity of the study.

However, it would be desired to offer a brief synopsis by mentioning the state-of-art of the application of the transcriptome approach to coffee, just mentioning some relevant previously literature:

Yuyama et al. (2016) Mol Genet Genomics, 291(1):323-36. doi: 10.1007/s00438-015-1111-x.
Cheng et al. (2017) Long-read sequencing of the coffee bean transcriptome reveals the diversity of full-length transcripts, GigaScience, Volume 6, Issue 11, November 2017, gix086, https://doi.org/10.1093/gigascience/gix086

Ivamoto et al. (2017). Transcriptome analysis of leaves, flowers and fruits perisperm of Coffea arabica L. reveals the differential expression of genes involved in raffinose biosynthesis. PLOS ONE, doi: 10.1371/journal.pone.0169595

- I would suggest to include ‘cellular totipotency’ in the Keywords.

- It is suggested that authors should add a reference to justify the choice of Cyclophilin as the reference gene, since it is not a commonly used gene;

- In line 146 describe better how these young leaves were sampled (how many and how much mass this represents);

- It is interesting to have the scale bars in Figure 1, at least for 1 B, 1C and 1E;

- In Figure 6 there is no indication whatsoever of a statistical analysis to compare expression levels.

Overall, the article should be accepted if the authors make some revisions, minor enough that I would NOT necessarily need to re-review it .

·

Basic reporting

The manuscript "Transcriptome analysis of ARF and Aux/IAA expression during the induction of somatic embryogenesis in Coffea canephora" by Quintana-Escobar et al. ((#36035) is a very interesting work concerning a transcriptomic analysis of somatic embryo induction and development in an economically relevant crop, Coffea canephora. The work is well designed and adequately developed. The authors have made use of up-to-date bioinformatic tools to obtain significant results about the implication of auxin signaling in the induction of somatic embryos in this species. Nonetheless, to my understanding, some changes should be made to publish it, as well as I would appreciate some questions to be clarified.
- English language: the text should be checked to change some incorrect English grammar expressions, some sentences that are not clear, as well as several typos and punctuation marks. I will give some examples below:
> English grammar expressions:
- The simple past of the verb "show" (showed) should be used instead of "show" or "shown". Examples can be found in Lines 203, 217, 222.
- Correct correlation between subject and verb. Line 278: "The expression changes of the gene correlates...".
- Adjectives have no plural: Line 163: "raws".
> Sentences that must be corrected or re-written:
- Line 68: "knowing in depth...". Line 215: "The recovery sequence rate...". Line 249: "...to the functional annotation as putative encodes to proteins involved...". Lines 404-405: the sentence should be rewritten.
> Typos and punctuation marks:
- Typos: Line 121: "This". Line 271: "ARAF6". Line 346:"AGAMUS". Line 398: "...in the account". Line 239: it should say "(14 DBI vs 21 DAI)".
- Punctuation marks: many commas are found in the text that should not be there, as well as they are missing in other positions. Besides, particles are also missing throughout the text. Just a couple of examples: Line 87: " The PGR mainly used are auxins, alone or in combination with other regulators, and...". Line 116: "Using a RNA-seq approach, it has been...".

- Scientific comments. Please consider the following suggestions:
- Expression of several genes is measured both with the RNA-seq data and with RT-PCR, but there is a lack of correlation between the values found. Were the same samples used for the two experiments? This should be addressed or clarified. For example, for the gene ARF18 the results are strikingly different in Fig 3, Fig 6 and Table 2. This should be explained. Indeed, the relative level of expression found for this gene with RT-PCR is extraordinary big for a transcription factor.
- Some interesting results are not discussed. Though the paper is focused on ARF and Aux/IAA genes, the identification of other genes relevant for the process is not discussed, and I think that they should be at least outlined in the Discussion. Particularly for the NAC protein, mitogen-activated kinase and the ABC transporter.
- At some points the Discussion lacks proper referencing. Examples can be found in Lines 330-335, 344-347 and 359-365.
- Lines 203-208: Do all the embryos generated successfully develop into a seedling?
- Discussion about ARF genes should be focused on the orthologs of the genes analyzed in this work, rather than a generalist discussion about ARFs.
- Line 105: I think that the genomic resources are characterized rather than generated.
- Lines 117-121: Do the species that are mentioned correspond to the references that follow?
- Some Figure legends are too brief, it would be interesting if they included more information.
- Why was the cyclophilin gene chosen as a reference gene for the RT-PCR experiments?
- Lines 376-379: It should say that those expression data are for SE in Arabidopsis.

In conclusion, this is a very interesting work, but some English language editing is needed. On the other hand, a discussion more focused on the results and a proper referencing would help increase the quality of the manuscript.

Experimental design

No comment.

Validity of the findings

No comment.

---

## Round 0.2 · Minor Revisions

Both reviewers agree that your revised manuscript responds adequately to their comments and confirm the scientific validity of your study. You might consider addressing their additional comments/ corrections in your final draft for production.

For the acceptance of the paper it would be important to provide the supplemental data as the following comments from a Section Editor suggests:

“This was a well designed manuscript in its presentation. It clearly defined a needed area of study, described the system, and designed an experimental body of work to generate data for study, and a series of observations were made to describe the results. However, on the other hand, for the reader to attempt validation of the experiment there is little to find in the context of the manuscript and there are no supplemental files. There are eight ARF and seven Aux/IAA genes highlighted, but no sequence data; only listed by the CcNN_gNNNNN name. Orthologs of Arabidopsis are pointed to, but not those of Coffea. The entire expression profile points to 25574 genes, but there is no reference to any of the raw data or SRA data resource. There is mention of a Coffea genome sequence, but no mention of how to access it. There is a simple statement indicating that Supplemental data is available online, but there is no direction or context on what is available. Since the manuscript concentrates on a fine description of gene expression in Coffea development, and that twelve libraries were created and from different tissues this study dictates that additional annotations be tied to the highlighted sequences; which would include gene ontology (GO) annotations be applied to the data to differentiate the members based on molecular function, biological process, and cellular component terms. This is a well developed manuscript with the exception that there is NO TRACTABLE DATA. If it can be supplied with supplemental files this would improve the revision status; however, an additional component is needed. Journal manuscripts are often scanned by text-mining software that locates and extracts core data elements, like gene function. Adding standard ontology terms, such as the Gene Ontology (GO, geneontology.org) or others from the OBO foundry (obofoundry.org) can enhance the recognition of your contribution and description. This will also make human curation of literature easier and more accurate. None of this was visible. I will place this manuscript in a “minor revision” status until such changes can be added to the future revision. I think this is a very worthy manuscript once data is provided to provide the reader context.”

Reviewer 1 ·

Basic reporting

The authors did good revisions. However there are still several problems awaiting further clarification.
1. For the 1st question in last revision, it's nice that a new figure has been inserted and there were a few clarifications. But, why 9DBI, 1DAI, 2DAI, and 21DAI were chosen needs more rationales.
2. For the 2nd question, the authors replied in the letter "the most dramatic changes in gene expression occurred during the first days after placing the explants in a new culture media with a different composition to begin the induction of SE". They also cited Cao et al. 2007 in the revised manuscript to support their argument. However, if i didn't misundertand it, did 9DBI relate to the main purpose of the study? If the main focus is to study gene expression change after explants were placed in new media, then there should be more transcriptome samples during the first few days, ie. between 14DBI and 9DBI? Otherwise 9DBI may not be necessary to the study. Please clarify that this point (also relates to the above question in terms of selection of the days of sampling).
3. For the 3rd question, the authors said they forgot this important criterion in the original version of the manuscript. As so there should be original results with p-values and fold change in the supplementary materials or by online links.

Experimental design

1. For the 1st question, i think 0.7 was reported as the minimum correlation coefficient between rna-seq and microarray in Sekhon et al. (2013). For Mukaka et al. (2012) I don't think it relates to rna-seq sample replicates directly. so I still am not convinced by the authors that their correlation between technical replicates is sufficient.

Validity of the findings

good

·

Basic reporting

The authors have made several changes to the manuscript that greatly improve its quality. The have carefully answered all reviewer´s suggestions, taking them into account to upgrade this research article. Furthermore, they have included a new Figure for clarity purposes. I believe that with these changes the work is significantly improved.

Minor comments or typos:
- Line 179: RStudio.
- Line 343: identified.
- Line 440: PGR.
- LIne 451: mitogen-activated protein kinase...

Experimental design

No comment.

Validity of the findings

No comment.

---

## Round 0.3 · accepted · Accept

Thank you very much for this second revision; especially for providing the FASTA sequences of ARF and Aux/IAA genes present in Coffea canephora.